# *Allium*-Derived Compound Propyl Propane Thiosulfonate (PTSO) Attenuates Metabolic Alterations in Mice Fed a High-Fat Diet through Its Anti-Inflammatory and Prebiotic Properties

**DOI:** 10.3390/nu13082595

**Published:** 2021-07-28

**Authors:** Teresa Vezza, José Garrido-Mesa, Patricia Diez-Echave, Laura Hidalgo-García, Antonio J. Ruiz-Malagón, Federico García, Manuel Sánchez, Marta Toral, Miguel Romero, Juan Duarte, Enrique Guillamón, Alberto Baños, Rocío Morón, Julio Gálvez, Alba Rodríguez-Nogales, María Elena Rodríguez-Cabezas

**Affiliations:** 1Center for Biomedical Research (CIBM), Department of Pharmacology, University of Granada, 18071 Granada, Spain; teresavezza@hotmail.it (T.V.); josegarridomesa@gmail.com (J.G.-M.); pdiezechave@gmail.com (P.D.-E.); lhidgar@gmail.com (L.H.-G.); a.jesus.ruiz14@gmail.com (A.J.R.-M.); manuelsanchezsantos@ugr.es (M.S.); martitj@hotmail.com (M.T.); miguelr@ugr.es (M.R.); jmduarte@ugr.es (J.D.); albarn@ugr.es (A.R.-N.); merodri@ugr.es (M.E.R.-C.); 2Instituto de Investigación Biosanitaria de Granada (ibs.GRANADA), 18012 Granada, Spain; fegarcia@ugr.es; 3Servicio Microbiología, Hospital Universitario Clínico San Cecilio, 18100 Granada, Spain; 4Centro de Investigación Biomédica en Red de Enfermedades Cardiovasculares (CIBERCV), 18001 Granada, Spain; 5DOMCA Innovative Food Solutions, DMC Research Center, 18620 Granada, Spain; eguillamon@domca.com (E.G.); abarjona@domca.com (A.B.); 6Servicio Farmacia Hospitalaria, Hospital Universitario Clínico San Cecilio, 18100 Granada, Spain; 7Centro de Investigaciones Biomédicas en Red–Enfermedades Hepáticas y Digestivas (CIBER-EHD), Department of Pharmacology, Center for Biomedical Research (CIBM), University of Granada, Avenida del Conocimiento s/n, 18100 Granada, Spain; 8Servicio de Digestivo, Hospital Universitario Virgen de las Nieves, 18012 Granada, Spain

**Keywords:** cytokines, dysbiosis, glucose metabolism, lipid metabolism, microbiota, obesity, organosulfur compound

## Abstract

Background: Propyl propane thiosulfonate (PTSO) is an organosulfur compound from *Allium* spp. that has shown interesting antimicrobial properties and immunomodulatory effects in different experimental models. In this sense, our aim was to evaluate its effect on an experimental model of obesity, focusing on inflammatory and metabolic markers and the gut microbiota. Methods and results: Mice were fed a high-fat diet and orally treated with different doses of PTSO (0.1, 0.5 and 1 mg/kg/day) for 5 weeks. PTSO lessened the weight gain and improved the plasma markers associated with glucose and lipid metabolisms. PTSO also attenuated obesity-associated systemic inflammation, reducing the immune cell infiltration and, thus, the expression of pro-inflammatory cytokines in adipose and hepatic tissues *(Il-1**ẞ*, *Il-6*, *Tnf-α*, *Mcp-1*, *Jnk-1*, *Jnk-2*, *Leptin*, *Leptin R*, *Adiponectin*, *Ampk*, *Ppar-α*, *Ppar-γ*, *Glut-4 and Tlr-4)* and improving the expression of different key elements for gut barrier integrity (*Muc-2*, *Muc-3*, *Occludin*, *Zo-1* and *Tff-3*). Additionally, these effects were connected to a regulation of the gut microbiome, which was altered by the high-fat diet. Conclusion: *Allium*-derived PTSO can be considered a potential new tool for the treatment of metabolic syndrome.

## 1. Introduction

*Allium* vegetables are well recognized for their beneficial properties from time immemorial. Garlic (*Allium sativum* L.) and onion (*Allium cepa* L.) have historical importance in the folk medicine of different cultures all over the world for treating heart problems, headache, colds, tumors and other ailments, as well as for boosting vigor [1]. These properties are attributed to its richness in minerals, essential amino acids and diverse phenolic and sulfur compounds [2]. Specifically, garlic is an excellent source of beneficial minerals, such as selenium, while onions are rich in flavonoids, such as quercetin, which is widely known for its biological properties [3]. Nevertheless, the most important bioactive compounds in the *Alliaceae* family are organosulfur compounds, mainly allyl cysteine derivatives, S-alk(en)yl-L-cysteine sulfoxides, thiosulfinates, thiosulfonates and sulfides, which are biosynthesized during tissue damage and confer useful biological properties, such as antimicrobial, anti-inflammatory, immunomodulatory, antioxidant, hepatoprotective and neuroprotective properties [3,4,5]. In fact, it is well described that these plants have cardioprotective effects that are associated with a positive effect on obesity and its associated metabolic disorders, including dyslipidemia, mild hypertension, hyperlipidemia, high blood glucose levels, impaired insulin sensitivity and liver lipotoxicity [6]. Thus, garlic was shown to reduce plasma lipid levels while increasing HDL cholesterol levels [7,8]. Moreover, it could also reduce the accumulation of fat in the first stages of atherosclerosis [9] and delay the calcification of the coronary arteries [10], improving their elasticity [11]. Moreover, it was reported that *Allium* compounds can inhibit the lipid accumulation and the transformation of monocytes into macrophages after stimulation with oxidized LDL cholesterol, which could explain the beneficial effects of garlic in models of atherosclerosis [12]), as well as enhance brown adipocyte-specific genes, like uncoupling protein 1, via the Krüppel-like factor 15 signal cascade [13]. These compounds also show anti-adipogenic effects by hindering 3T3-L1 adipocyte differentiation in vitro via the activation of AMP-activated protein kinase (AMPK) and carnitine palmitoyltransferase, as well as the inhibition of acetyl CoA carboxylase-1 [13], or by means of extracellular signal-regulated kinase activation [14]. Furthermore, a recent meta-analysis has proposed that garlic supplementation may decrease waist circumference without changing body weight or body mass index [15]. A parallel, double-blind, placebo-controlled, randomized study reported the ability of aged garlic supplementation to modulate immune cell distribution and decrease serum tumor necrosis factor (TNF)-α and interleukin (IL)-6 levels in healthy obese adults, thus reducing obesity-induced inflammation [16]. In this regard, garlic consumption has been reported to decrease resistin levels, a pro-inflammatory adipokine, in overweight and obese women with osteoarthritis and reduce the pain compared with placebo patients in a randomized, double-blind, placebo-controlled, parallel design trial [17]. 

Although many studies have been undertaken to examine the cardioprotective and anti-obesity effects of *Allium* and their organosulfur products, the outcomes are very variable; therefore, more investigations, both preclinical and clinical, are needed to better elucidate the effects of *Allium* products and establish their use and safety in preventing the metabolic syndrome.

In this sense, we evaluated an *Allium* organosulfur compound, namely, propyl propane thiosulfonate (PTSO), in an experimental model of high-fat-diet-induced metabolic syndrome in mice. This product does not present toxic effects [18] and already showed interesting antimicrobial properties for the livestock industry [19] and immunomodulatory effects in experimental colitis [20]. Thus, concerning the latter, it was shown to be able to downregulate pro-inflammatory mediators, improve the intestinal mucosal integrity and ameliorate colitis-associated dysbiosis [20]. These effects could also contribute to the amelioration of metabolic-syndrome-associated inflammation and its derived adverse sequels. The present study has assessed the effects of PTSO on the metabolic alterations and the inflammation that characterizes the metabolic syndrome, as well as on its abilities to modulate the gut microbiota.

## 2. Materials and Methods

### 2.1. Experimental Animals and Diets

The study was carried out following the “Guide for the care and use of laboratory animals” of the National Institute of Health (Washington, DC, USA), and the protocols were approved by the Committee of Ethics of the University of Granada (reference no. 28/03/2016/030). Five-week-old male C57Bl/6 mice (Charles River, Barcelona, Spain) were kept in the Animal Facility of the University of Granada at a controlled temperature and humidity (22 ± 1 °C, 55 ± 10% relative humidity) with a 12 h light/dark cycle and free access to food and drink. Mice were randomly divided into several groups (*n* = 10): lean (control diet), lean treated (control diet treated), obese (HFD) and obese treated (HFD treated). The lean mice were fed standard chow (210 Control Diet), while obese mice received a high-fat diet (HFD) in which 60% of the caloric content came from fat (Purified Diet 230 HF). The experimental design was as follows: obese treated mice were administered different doses of PTSO (0.1, 0.5 and 1 mg/kg/day v.o.) dissolved in water (100 μL), while lean treated mice received 1 mg/kg/day of PTSO under the same conditions. Lean (control diet) and obese (HFD) groups were administered the same amount of water daily. The mice were treated for 5 weeks and the weight and consumption of food and water were monitored twice a week. 

### 2.2. Glucose Tolerance Test and Plasma Determinations

At week 4, mice were fasted for 8 h and a glucose tolerance test was carried out as previously described [21] (see the Appendix A). At the end of the treatment, the animals were sacrificed after taking a blood sample via a cardiac puncture. The blood was centrifuged to separate the plasma at 5000× *g* for 20 min at 4 °C, which was frozen at −80 °C until the following determinations were made: glucose, LDL cholesterol, HDL cholesterol, total cholesterol and insulin (see the Appendix A).

### 2.3. Morphological Variables

After the sacrifice, the adipose tissues (epididymal and abdominal fat) were removed, cleaned and weighed. The relationship between fat and animal size was estimated by dividing the weight of the fat by the length of the tibia. The liver and colon were also removed, which were likewise cleaned and weighed. All samples were frozen in liquid nitrogen and stored at −80 °C until further processing.

### 2.4. Gene Expression Analysis Using RT-qPCR

The total RNA of the different tissues was extracted with NucleoZOL^®^ (Macherey-Nagel, Düren, Germany) according to the manufacturer’s instructions. A reverse transcription (RT) was then performed with oligo (dT) and reverse transcriptase primers M-MLV (Promega, Southampton, UK) in a TProfessional Basic Thermocycler (Biometra, Göttingen, Germany). The real-time polymerase chain reaction (qPCR) was performed in 48-well optical grade plates in EcoTM real-time PCR equipment (Illumina, San Diego, CA, USA) with 10 ng of complementary DNA, KAPA SYBR^®^ FAST qPCR Master Mix (Kapa Biosystems, Wilmington, MA, USA) and specific oligonucleotides at their hybridization temperature (Appendix A). To normalize messenger RNA expression, the housekeeping gene glyceraldehyde 3-phosphate dehydrogenase (*Gapdh*) was measured. The relative quantification of messenger RNA was estimated using the ΔΔCt method.

### 2.5. DNA Extraction and Illumina MiSeq Sequencing

DNA from fecal contents was extracted as described by Rodríguez-Nogales et al. [22]. The resulting sequences were quality-filtered, clustered and taxonomically assigned on the basis of a 97% similarity level against the SILVA database [23] using the QIIME software package (version 1.9.1) (Knight Lab, San Diego, CA, USA). Sequences were chosen to estimate the total bacterial diversity of the DNA samples in a comparable manner and were trimmed to clear away barcodes, primers, chimeras, plasmids, mitochondrial DNA and any non-16S bacterial reads and sequences that were <150 bp in size.

### 2.6. Vascular Reactivity Studies and NADPH Oxidase Activity

Descending thoracic aortic rings were dissected and the isometric tension was measured as described before [24]. Briefly, the aortic rings were placed in an organ chamber filled with Krebs solution (composition in mM: NaCl 118, KCl 4.75, NaHCO_3_ 25, MgSO_4_ 1.2, CaCl_2_, KH_2_PO_4_ 1.2 and glucose 11) at 37 °C and gassed with 95% O_2_ and 5% CO_2_ (pH 7.4), suspended in a wire myograph (model 610M, Danish Myo Technology, Aarhus, Denmark) and loaded with a tension of 5 nN. After a 90 min stabilization period, cumulative concentration–response curves to acetylcholine (10^−9^–10^−5^ M) were carried out in intact rings pre-contracted by U46619 (10^−8^ M). Relaxant responses to acetylcholine were expressed as a percentage of precontraction. The length–tension relationship was calculated with the myograph software (Myodaq 2.01, Danish Myotechnologies, Denmark).

NADPH oxidase activity in intact aortic rings was evaluated with the lucigenin-enhanced chemiluminescence assay, as reported before [25] (see the Appendix A for details). 

### 2.7. Flow Cytometry

The cells from adipose and liver tissue were isolated following the procedure previously reported, with some modifications [26] (see the Appendix A). 

### 2.8. Statistics

All results are expressed as mean ± standard error of the mean. Statistical significance between the different groups was calculated with a one-way analysis of variance (ANOVA) and post hoc tests of significance. Differences between proportions were evaluated with chi-square analysis. All statistical analyses were performed with the GraphPad 8 program (GraphPad Software, Inc., La Jolla, CA, USA), establishing the significant differences at *p* < 0.05.

## 3. Results and Discussion 

Metabolic syndrome is defined by WHO as a pathological condition in which abdominal obesity, insulin resistance, hypertension and hyperlipidemia may concur. Thus, individuals suffering from it have an increased risk of cardiovascular mortality and morbidity [27].

Two main drivers contribute to the advance of this disease, namely high-calorie–low-fiber-food consumption and low physical activity; therefore, the first action that needs to be taken is the promotion of lifestyle changes. However, when these actions are not enough to control the symptoms, a pharmacological approach is necessary to prevent the associated long-term effects. Metabolic syndrome is quite complex; consequently, the long-term pharmacological interventions may include the use of several drugs to treat the different complications, though they can also produce adverse effects [28] or become ineffective in the long run.

Therefore, the search for new alternative and safe products that holistically treat the associated symptoms of metabolic syndrome has become a priority. Garlic, onion and their organosulfur compounds show potential, but more work is needed to consider their use. In this study, we explored the effects of PTSO, an *Allium* component with antioxidant, anti-inflammatory and immunomodulatory properties [20], in a metabolic syndrome mouse model that was induced by a high-fat diet (HFD).

### 3.1. PTSO Treatment Reduced Body Weight Gain and Ameliorated Metabolic Alterations in HFD-Fed Mice 

The mice that consumed an HFD experienced a significantly greater weight gain than the mice fed the standard diet. However, HFD mice treated with PTSO (HFD-PTSO) experienced significantly lower weight gain compared to non-treated HFD mice (Figure 1A). This was not derived from a satiating effect since the treatment did not affect energy intake. However, it significantly reduced energy efficiency (Figure 1A). Moreover, a significantly greater accumulation of adipose tissue was detected in the untreated HFD mice compared to control diet mice, while all doses of PTSO significantly diminished it (Figure 1B). 

The glucose tolerance test showed similar curves for all groups, reaching the glucose peak at 15 min and decreasing to baseline values at 120 min (Figure 1C). However, blood glucose values, both fasting and during the test, were over 300% higher in the HFD mice. Thus, the area under the curve of obese mice had significantly higher values than non-obese mice. The treatment significantly reduced this parameter by at least 20%, with no dose–response relationship, in comparison with the obese non-treated mice, although the value was still significantly different from non-obese mice (Figure 1C). The evaluation of glucose homeostasis also included the measurement of the fasting plasma insulin level (Figure 2A). The high-fat diet had no effect on this parameter, but the treatment with PTSO at 0.5 and 1 mg/kg decreased it significantly. Interestingly, when the marker of insulin resistance HOMA-IR was calculated after taking into account the plasma glucose and insulin values, it appeared significantly elevated in obese mice compared to the lean ones (Figure 2A). The treatment with the two highest doses of PTSO reduced it significantly, although they were still different from the lean mice. Therefore, the PTSO treatment was found to improve the glucose metabolism, reducing the plasmatic levels and the systemic intolerance and insulin resistance, after considering the HOMA-IR, in mice fed with the HFD, which agrees with what was previously described for garlic, onion and their derivative [16,29,30,31].

Moreover, in addition to having greater adipose tissue deposits, the obese mice showed an alteration in the plasma lipid profile, which was characterized by an elevation in the total, HDL and LDL cholesterol levels, as well as an alteration in the LDL/HDL cholesterol ratio, compared to lean mice. The PTSO also improved the lipid metabolism, reducing the total and LDL cholesterol without modifying the HDL cholesterol (Figure 2B), as it has been reported for aged garlic supplementation in obese adult patients [16].

### 3.2. PTSO Treatment Lessened Inflammation and Improved Gut Barrier in HFD-Fed Mice

The onset and progression of these metabolic alterations are intimately related to a chronic low-grade systemic inflammation that is typified by immune cell infiltration in the metabolic tissues, liver and fat, and the subsequent overproduction of chemokines and cytokines, as we observed. In this sense, mice fed the HFD showed increased gene expression of pro-inflammatory mediators in the liver (*Tnf-α*, *Il-1β*, *Il-6* and the attractant monocytes chemotactic protein 1 (*Mcp-1*)) and in adipose tissue (*Tnf-α* and *Il-6*) (Figure 3). As reported before in a model of experimental colitis [20,32,33], PTSO hindered the over-expression of the pro-inflammatory mediators evaluated, including *Il-1β* and *Mcp-1* in the liver, and *Tnf-α* and *Il-6* both in the liver and fat, which was linked with impaired insulin signaling. Previous studies also described the anti-inflammatory properties of aged garlic extract and *Allicin*, which were found to inhibit the production of nitric oxide, TNF-α and IL-4, and could support their application for atherosclerotic vascular disease [33]. The JNK pathway is considered a stressor sensor since JNK signaling can modulate cytokine synthesis, as well as be activated by these cytokines. This is important for keeping homeostasis, but when there is deregulation, JNK activation generates an aberrant production of cytokines that leads to chronic inflammation and the development of metabolic disorders [34]. In fact, JNK signaling has been linked with cardiometabolic inflammation, with the infiltration of immune cells into liver and fat tissues being JNK-dependent [35]. Accordingly, obese mice showed a significant over-expression of *Jnk-1* and *Jnk-2* in liver and fat tissues (Figure 3). However, treatment with PTSO normalized them in both tissues, thus downregulating the obesity-associated inflammatory systemic response. AMPK is another nutrient sensor that, unlike JNK, inhibits inflammation and oxidative stress, with its inactivation being linked to the pathogenesis of metabolic syndrome and associated conditions [36]. The control HFD-fed mice displayed a reduced expression of *Ampk*, while the PTSO treatment reverted it, both in the liver and fat. This is very interesting since widely used antidiabetic drugs, including metformin and rosiglitazone, act as insulin sensitizers through AMPK activation [37].

As expected, the expression of the adipokines *Leptin* (in fat tissue) and *Adiponectin* (in fat tissue and liver) were modified in obese mice, in association with a reduced expression of *Leptin Receptor (Leptin R)* in both liver and adipose tissues. PTSO had no effect on *Leptin* expression, but it significantly augmented the expression of its receptor in both tissues, thus indicating a partial amelioration of the obesity-associated dysfunction in leptin-mediated signaling (Figure 4A). Nevertheless, it normalized the expression of *Adiponectin*, both in the liver and fat. This effect may contribute to the improvement of insulin sensitivity and glucose and fat metabolisms [38,39]. Moreover, adiponectin was reported to show anti-inflammatory and antioxidant activities that could mediate the effects of PTSO [40].

PPARs were described to exert a prominent role in obesity and inflammation. Thus, PPARα appears expressed in metabolically active tissues, including fat and the liver, as well as in immune cells [41]. PPARα was reported to exert anti-inflammatory effects in fat tissue, which was mediated via different mechanisms that include decreasing adipocyte hypertrophy and inhibiting inflammatory genes [42]. Its expression was downregulated in control obese mice, while the treatment with PTSO increased it (Figure 4B). PPARγ was also described as an inhibitor of pro-inflammatory gene expression by reducing macrophage infiltration and upregulating the expression of adiponectin [43]. Thus, the control obese mice displayed a decreased expression in their fat tissue, which, interestingly, was normalized by the PTSO treatment. Of note, the treatment of obese rats with troglitazone, a synthetic PPARγ agonist, was able to considerably diminish the adipocytes, as well as the expression of TNF-α in comparison with untreated rats [44]. Thus, these effects on PPARs could participate in the anti-inflammatory effect demonstrated by the garlic compound. 

As commented before, the metabolic disorders linked to obesity are intimately related to the inflammatory status that is developed. In vitro studies have reported that IL-6 reduces the expression of adiponectin, glucose transporter-4 (GLUT-4) and insulin receptor substrate-1 (IRS-1), while TNF-α causes the elevated secretion of MCP-1 and IL-6 from pre-adipocytes [45], which agrees with the results presented in this study. In fact, IL-6 overproduction was associated with reduced GLUT-4 expression [46], as it was also observed in the obese mice in the current study (Figure 5A). In adipocytes, glucose uptake in normal conditions takes place via insulin-stimulated transport, which is mainly mediated by GLUT-4 [47], but when there is an excess of glucose in the blood, this is diffused into adipocytes through GLUT-4, triggering the synthesis of fatty acids and glycerol and inhibiting lipolysis. Nevertheless, *Glut-4* gene expression in adipose tissue is hindered in obesity-associated insulin resistance [48]. Consequently, this could participate in insulin resistance and produce the increased plasma glucose levels that were detected. Remarkably, the PTSO treatment significantly upregulated *Glut-4* expression (Figure 5A), which may have contributed to enhancing the insulin sensitivity and thus blood glucose uptake into adipocytes, as well as producing lower blood glucose levels.

The pro-inflammatory and oxidant environment that characterizes obesity may be triggered by so-called “metabolic endotoxemia,” which is associated with a low-level increase of gut-derived lipopolysaccharide (LPS) that acts through TLR4 [48]. Accordingly, we observed significantly higher levels of circulating LPS in obese mice, together with a greater expression of *Tlr-4* (Figure 5A), which agrees with previous reports. In this regard, we showed that HFD can increase paracellular transport of bacterial products by impairing intestinal permeability since we evidenced a reduced colonic expression of tight junction proteins, *Occludin*, *Zo-1* and *Tff-3*, as well as mucins (*Muc-2* and *-3*) (Figure 6A), in which an altered intestinal microbiota composition can have a key role [49]. Extraordinarily, PTSO reduced the plasma LPS levels and *Tlr-4* expression in the liver, as well as normalized the expression of these key elements for the epithelial gut barrier integrity, as was previously seen in an experimental model of mouse colitis [20].

Bearing the above in mind, it is clear that the impact of PTSO treatment on the immune response may contribute to the positive effects, maybe by restoring the infiltration and composition of the immune population in fat and liver tissues. In this sense, it is well described that immune cells, such as myeloid-derived suppressor cells (MDSCs) (Ly6C^+^CD11b^+^), are a diverse subset of immature and mature myeloid cells with immunoregulatory properties [50,51] (Figure 5B). Under physiological conditions, immature myeloid cells (IMCs) differentiate into mature granulocytes, macrophages or dendritic cells, while in pathological conditions, such as in inflammatory diseases, the overproduction of pro-inflammatory mediators boosts the proliferation of IMCs and partially blocks their differentiation producing an accumulation of MDSC. In fact, the liver is the major organ where IMCs accumulate [52]. Our data correspond to data published by other authors [53], where the percentage of total MDSCs (Ly6C^+^CD11b^+^) in the liver was augmented in obese mice compared to control mice, which indicates a blockage in the regular differentiation of these cells and, therefore, an accumulation of these cells in the liver. Further studies are needed to understand why these cells are especially recruited into the liver. Several hepatic cell populations, such as hepatocytes, Kupffer cells, sinusoidal endothelial cells and hepatic satellite cells, are able to produce chemokines and/or chemotactic cytokines upon activation. These mediators may control the migration of these cells and drive their accumulation in the liver. HFD-derived compounds could trigger these cells in the liver and, thus, stimulate the recruitment of these circulating activated immature myeloid cells into the liver. Among these mediators, IL-6 could be highlighted. Studies showed that IL-6 is over-expressed in NAFLD patients and it was also reported to inhibit immature myeloid cell differentiation [54,55]. Subsequently, these activated cells are accumulated in the liver. This hypothesis was confirmed by our results (Figure 5B), where *Il-6* expression was also increased in liver tissue. Remarkably, PTSO treatment restored this population to normal values, along with its *Il-6* expression levels. Nevertheless, macrophages and dendritic cells are widely known as key regulators of this inflammatory process. These cells can be reshaped and respond to varied stimuli, including metabolic signals [56]. It is widely described that in the liver and adipose tissue, the accumulation of inflammatory macrophages and dendritic cells contributes to the deregulation of glucose homeostasis, increase of obesity-induced inflammation and hepatic fibrosis [53,57,58]. In the present study, we confirmed these previous studies (Figure 5B). Hepatic pro-inflammatory macrophages’ population and plasmatic glucose levels were increased in obese mice and, interestingly, the treatment was able to restore both determinations. Moreover, these immune cells were also pivotal in the fat tissue, where the existence of a substantial population of pro-inflammatory macrophages and DCs in the HFD group was significantly re-established by the treatment (Figure 5B). These results confirm the inhibitory effect exerted by PTSO against macrophage/DC cell infiltration and, therefore, an improvement in the inflammatory status characteristic of this pathological condition.

### 3.3. PTSO Treatment Improved Endothelial Dysfunction

Another manifestation of the inflammatory process associated with HFD-induced obesity in mice is an endothelial dysfunction characterized by an altered response to acetylcholine, which was previously reported by other authors. This encompasses the vascular synthesis of reactive oxygen species (ROS), including superoxide anion, which quickly inactivates nitric oxide (NO) [53], the main source of vascular superoxide anion in obese rodents [59] and humans [60]. The vasorelaxant effect of acetylcholine on endothelium-intact aortic rings was significantly reduced in HFD mice in comparison with the control diet group (Figure 6B), with the values of the maximal relaxant response in HFD mice being lower than in the control diet mice, although no significant changes were observed in the concentration of acetylcholine that produced the half-maximal relaxation (Figure 6B). PTSO treatment given to HFD-fed mice improved endothelial function since it ameliorated the endothelium-dependent relaxation induced by acetylcholine (Figure 6B). Agreeing with the literature, non-treated obese mice presented altered vasodilatation in response to acetylcholine mediated by an increased NAPDH oxidase activity, evidencing an endothelial dysfunction (Figure 6B). However, the treatment with PTSO normalized the NADPH activity and, consequently, improved endothelial dysfunction (Figure 6B). This may justify the traditional use of garlic and its derivatives for the treatment of hypertension and other cardiovascular conditions, which needs more studies to determine their true impact, as supported by the latest meta-analyses [61,62].

### 3.4. Prebiotic Properties of PTSO Modulate Gut Dysbiosis in HFD-Fed Mice 

As mentioned above, obesity is linked to an alteration of the intestinal microbiota composition, which is induced by the consumption of a high fat/energy diet that leads to metabolic endotoxemia and thus low-grade systemic inflammation and metabolic alterations. This obesity-associated dysbiosis, mostly triggered by external factors, can produce a strong alteration of the symbiotic relationship between the gut microbiome and the host. Thus, it can prompt the development of metabolic diseases [63]. In this sense, both in humans and mice studies, it is well described that the enrichment of *Firmicutes*, together with a reduction in *Bacteroidetes*, is associated with these disorders. Administration of PTSO was able to counteract the altered composition and the diversity in the gut microbiota, normalizing the proportion of the major bacteria phyla seen in standard diet-fed mice (Figure 7A,B; Appendix A). This demonstrates that the change in the gut bacterial composition by the PTSO treatment is related to amelioration in obesity-associated dysbiosis. Since the *Firmicutes*-enriched microbiome demonstrated enhanced energy harvesting from food [64], the relative underrepresentation of *Firmicutes* in PTSO-HFD mice could lessen energy assimilation and potentially contribute to the observed resistance to diet-induced obesity. Additionally, the significant modifications observed in control HFD-fed mice in the proportions of *Bacteroidetes* and *Verrumicrobia* were also restored in those obese mice treated with PTSO (Figure 7B). Furthermore, it is important to highlight the role of *Akkermansia muciniphila* in obesity, which was considered as the dominant human bacterium that abundantly colonizes this nutrient-rich environment [65]. In fact, it was reported that the abundance of this bacterial species is inversely associated with body weight and type 1 diabetes in mice and humans [66,67]. Actually, *A. muciniphila* treatment was reported to reverse HFD-induced metabolic disorders [68]. In this study, a reduction in the proportion of the genus *Akkermansia* was seen in control obese mice, and this was overturned by treatment with PTSO. The restoration of the abundance of *Akkermansia* sp. exerted by PTSO could be linked to the improvement of the gut barrier function through the enlarged production of mucins in the colonic tissue, considering that they are the principal nutrients for these bacteria. 

Additionally, *Lachnospiraceae* bacteria were associated with obesity and diabetes [69,70]. A metagenomic study reported that the taxonomic family *Lachnospiraceae* may be associated with type 2 diabetes in humans and mouse models [70,71]. Moreover, *Lachnospiraceae* is positively linked to the plasma fasting glucose concentration, aspartate transaminase activity, insulin signaling in response to nutrient availability via the mammalian target of rapamycin and contributes to the development of diabetes [69]. This pattern agrees with a previous study that was carried out in mice, indicating the importance of the family *Lachnospiraceae* for metabolic diseases, such as obesity and diabetes [69]. Our present data clearly showed that HFD highly increased the abundance of *Lachnospiraceae*, which is positively associated with the increased inflammatory status, while PTSO treatment was able to restore it and improved the inflammatory status (Figure 7C). Another *Allium* product, namely, alliin, has also been reported to reduce *Lachnospiraceae* [31]. Therefore, this suggests that the possible mechanisms involved in regulating glucose metabolism due to PTSO treatment can be associated with the reduction of *Lachnospiraceae* in the gut.

The other important result obtained in this study concerning the gut microbiota was that HFD also significantly amplified the relative abundance of *Streptococcaceae.* This finding brings to light the relationship between obesity-related inflammatory bowel diseases and colon cancer because *Streptococcaceae* was connected to metabolic syndrome and colon cancer [72,73]. The ability of PTSO administration to restore its abundance suggests that PTSO could also be considered for the development of a novel approach to prevent colorectal cancer (Figure 7C). 

Moreover, *Lactobacillus* was also described to have an anti-obese effect in diet-induced obesity murine models, and the probable mechanisms might concern the preservation of the intestinal barrier and protection from chronic inflammation [74]. These results were confirmed by our study, where the HFD group showed a reduced proportion of *Lactobacillus* compared with control mice. Remarkably, the PTSO treatment increased the *Lactobacillus* abundance and improved markers that are associated with the barrier integrity maintenance and the inflammatory process (Figure 7C).

## 4. Conclusions

In conclusion, *Allium*-derived PTSO was shown to globally treat the different conditions that are associated with metabolic syndrome in this experimental model of high-fat-diet-induced obesity in mice. The mechanisms behind its positive effects may be mediated by their antioxidant, anti-inflammatory and prebiotic activities, which eventually reduced the low-grade obesity-associated systemic inflammation and improved the glucose and lipid metabolisms. Therefore, PTSO constitutes a potential new tool for the management of metabolic syndrome.

## Figures and Tables

**Figure 1 nutrients-13-02595-f001:**
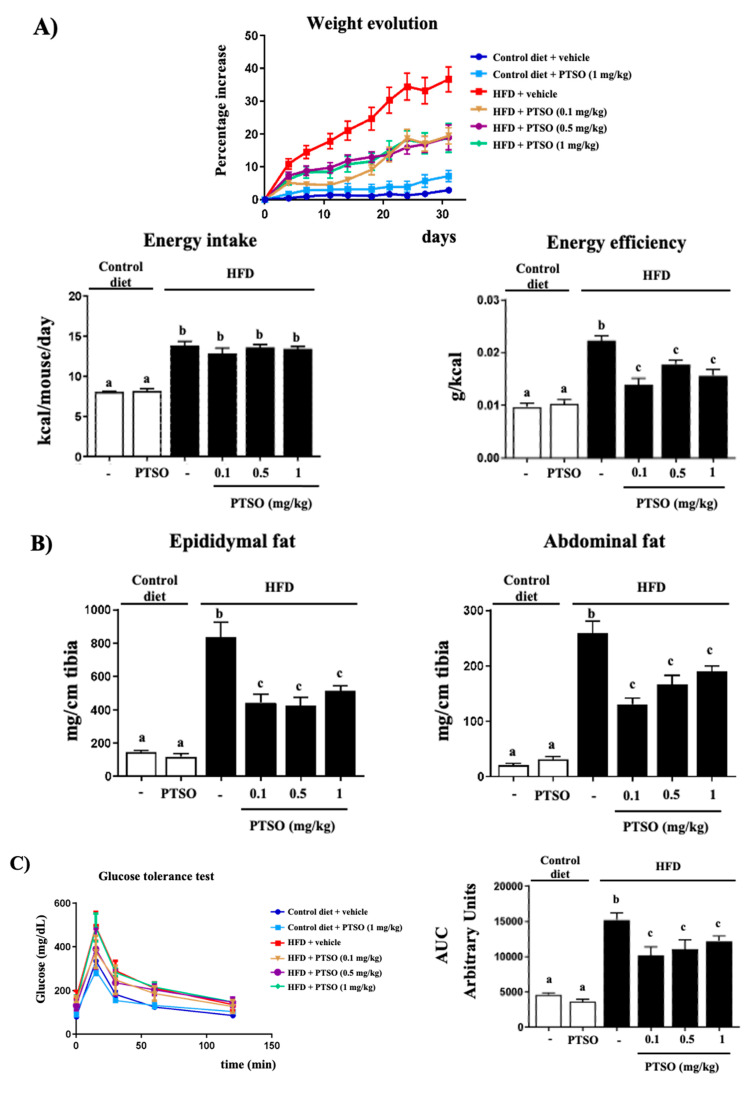
Effects of propyl propane thiosulfonate (PTSO) administration on (**A**) morphological changes (body weight evolution, energy efficiency and energy intake); (**B**) epididymal and abdominal fat; (**C**) glucose tolerance test and area under the curve (AUC) in the control and high-fat diet (HFD)-fed mice. Groups with different letters statistically differed (*p* < 0.05).

**Figure 2 nutrients-13-02595-f002:**
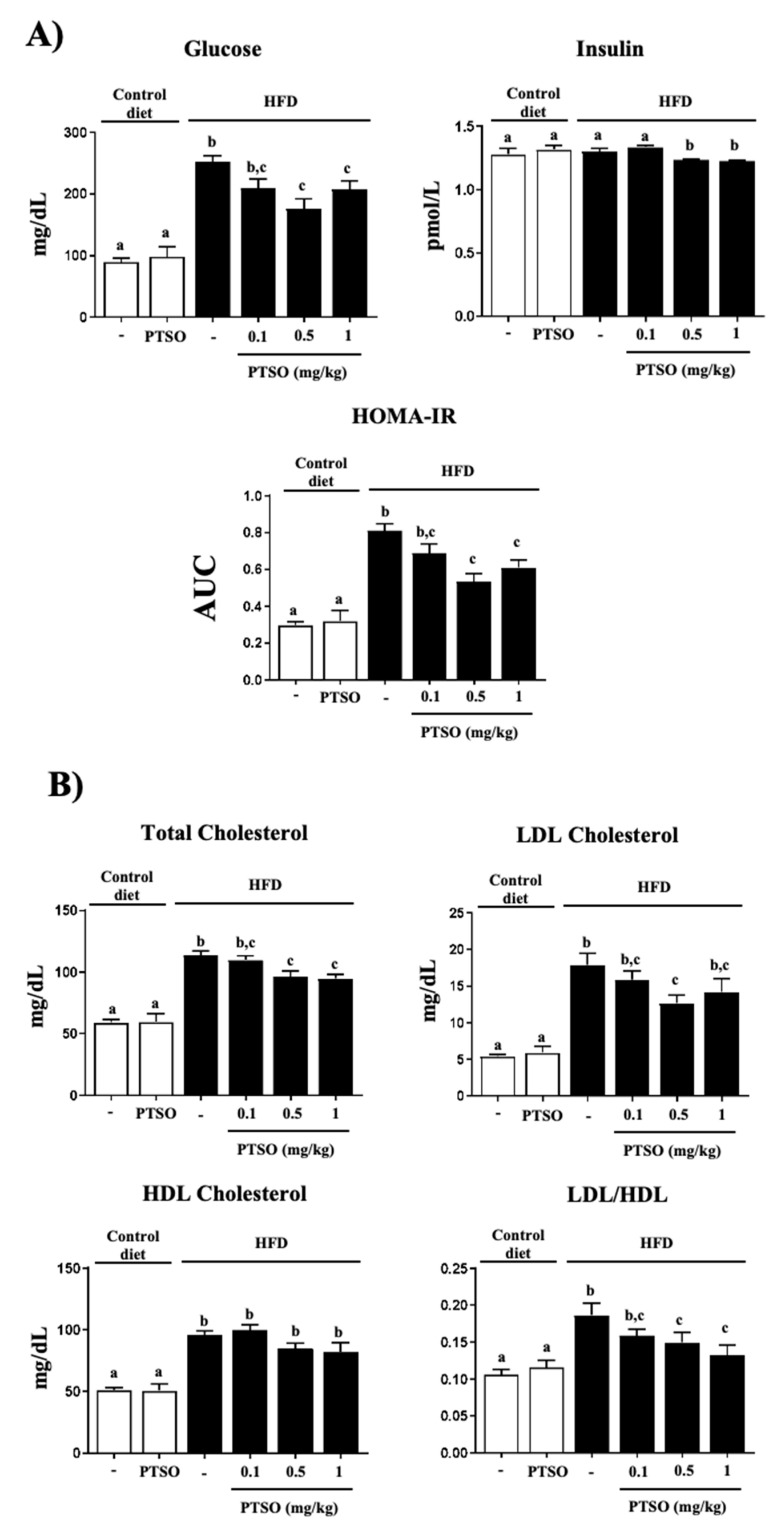
Effects of propyl propane thiosulfonate (PTSO) administration on (**A**) basal glucose, insulin levels and homeostasis model assessment insulin resistance (HOMA-IR) index; (**B**) total, LDL and HDL cholesterol plasma levels in the control and high-fat diet (HFD)-fed mice. Groups with different letters statistically differed (*p* < 0.05).

**Figure 3 nutrients-13-02595-f003:**
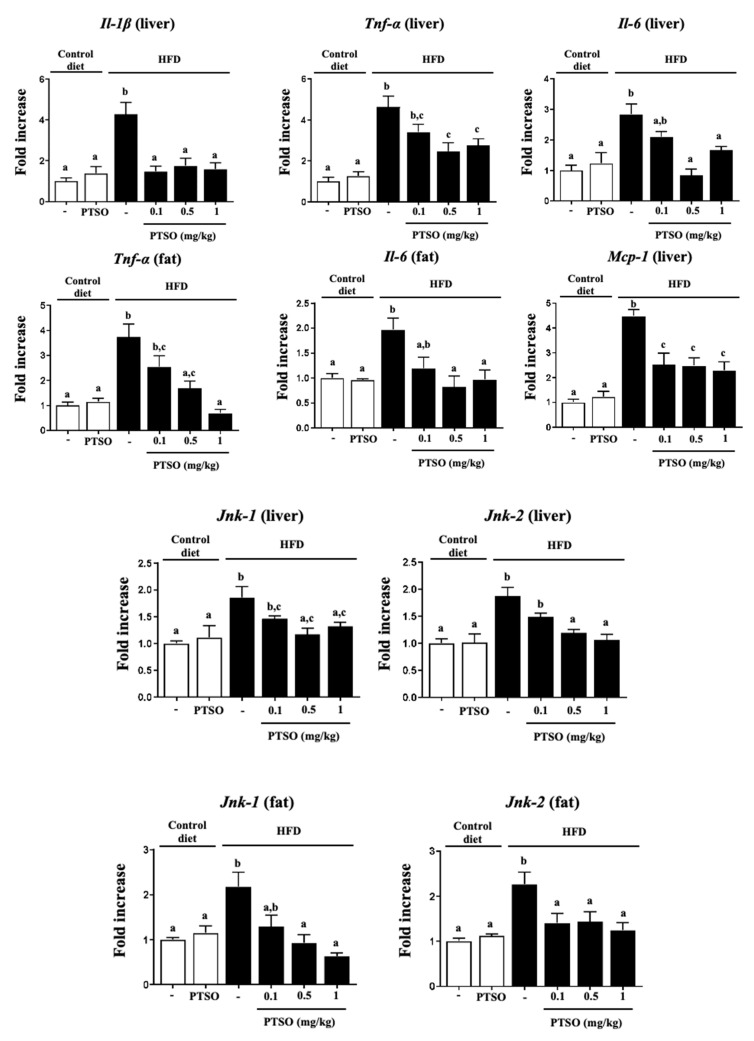
Effects of propyl propane thiosulfonate (PTSO) administration on the liver and fat gene expression of *Il-1β*, *Tnf-α*, *Il-6*, *Mcp-1*, *Jnk-1* and *Jnk-2* in control and high-fat diet (HFD)-fed mice, which was analyzed using real-time qPCR. Groups with different letters statistically differed (*p* < 0.05).

**Figure 4 nutrients-13-02595-f004:**
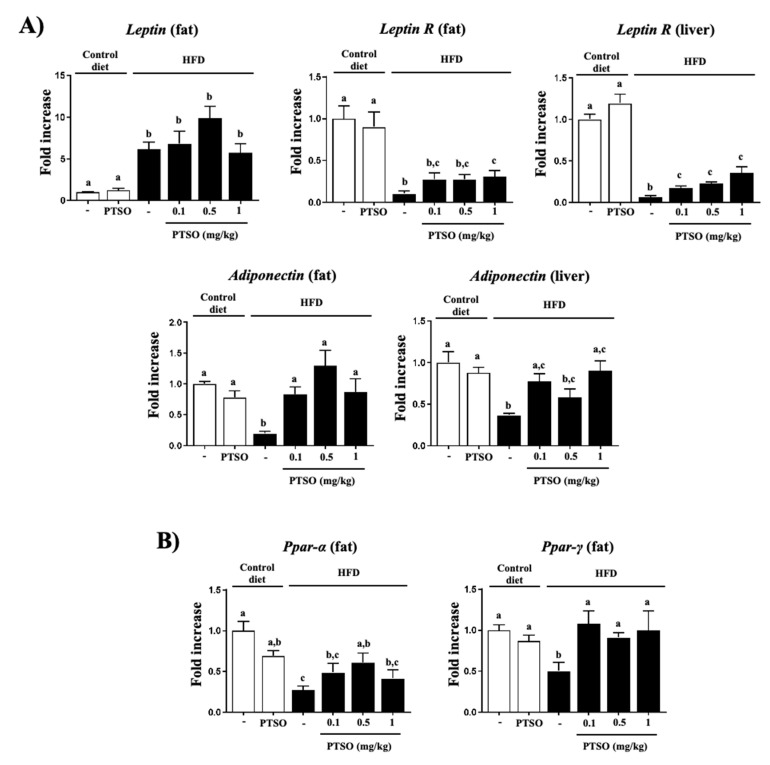
Effects of propyl propane thiosulfonate (PTSO) administration on (**A**) *Leptin*, *Leptin R* and *Adiponectin* expression in the liver and fat and (**B**) on fat and liver gene expression of, *Ppar-α* and *Ppar-γ* in control and high-fat diet (HFD)-fed mice. Analysis performed by real-time qPCR. Groups with different letters statistically differed (*p* < 0.05).

**Figure 5 nutrients-13-02595-f005:**
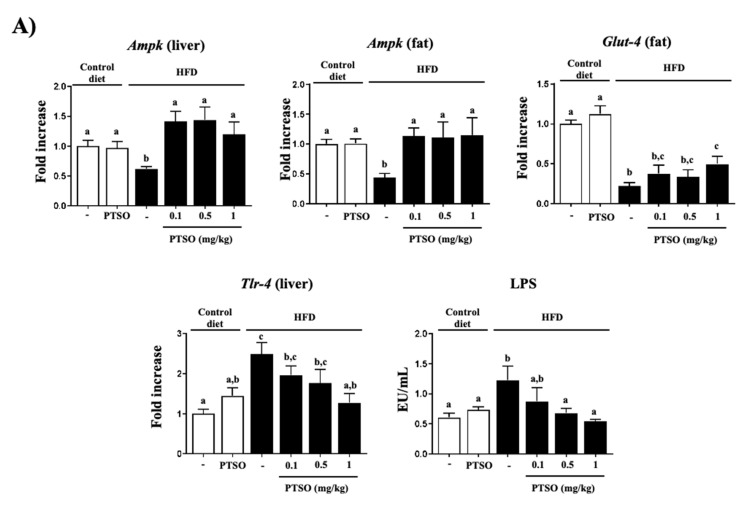
Effects of propyl propane thiosulfonate (PTSO) administration on the liver and fat: (**A**) gene expression of *Ampk*, *Glut4* and *Tlr4*, which was analyzed using real-time qPCR and LPS plasma concentrations; (**B**) infiltration and composition of the immune population in control and high-fat diet (HFD)-fed mice. Groups with different letters statistically differed (*p* < 0.05).

**Figure 6 nutrients-13-02595-f006:**
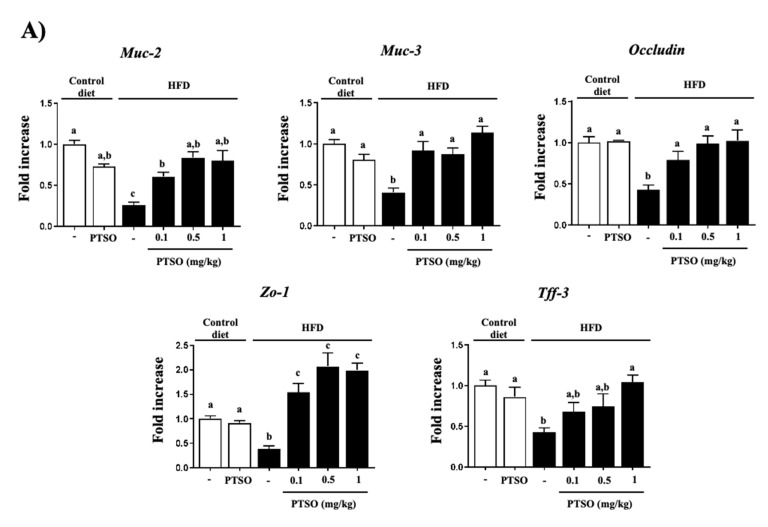
Effects of propyl propane thiosulfonate (PTSO) administration on (**A**) colonic gene expression of *Muc-2*, *Muc-3*, *Occludin*, *Zo-1* and *Tff-3*; (**B**) endothelium-dependent relaxation and aortic NADPH activity in control and high-fat diet (HFD)-fed mice. Groups with different letters statistically differed (*p* < 0.05).

**Figure 7 nutrients-13-02595-f007:**
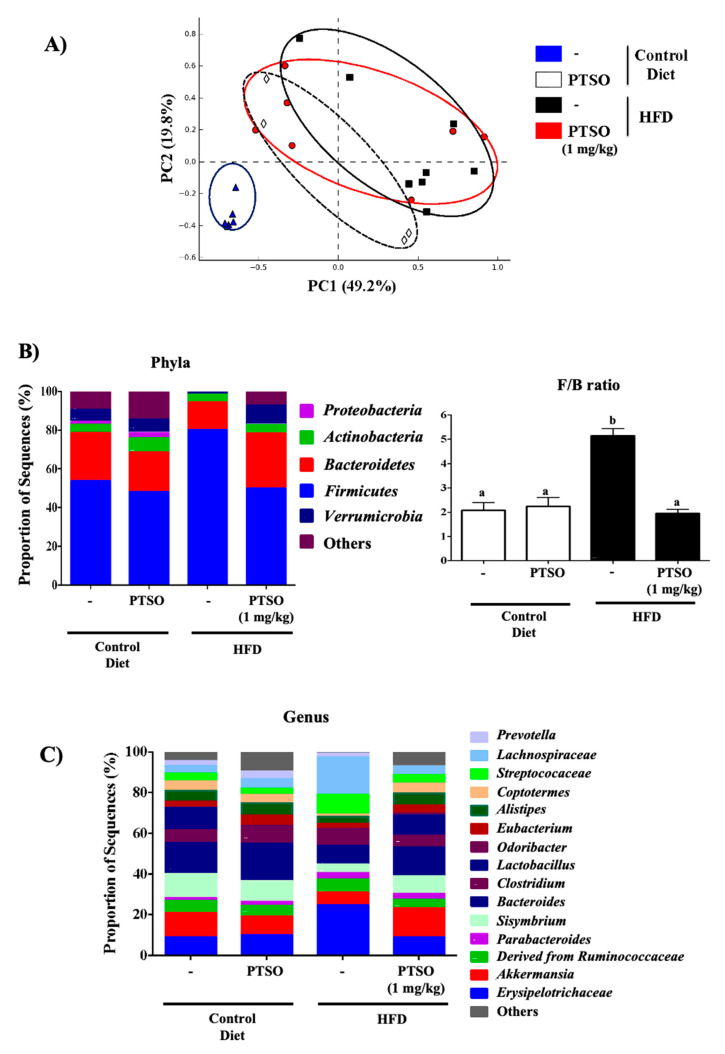
Impact of propyl propane thiosulfonate (PTSO) on fecal microbiota composition: (**A**) principal component analysis plot; (**B**,**C**) bacterial community (phyla, genus) and the *F/B* ratio. Groups with different letters statistically differed (*p* < 0.05).

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
