# Peer review of "Allium-Derived Compound Propyl Propane Thiosulfonate (PTSO) Attenuates Metabolic Alterations in Mice Fed a High-Fat Diet through Its Anti-Inflammatory and Prebiotic Properties"

_nutrients, 2021, doi:10.3390/nu13082595_

Round 1

Reviewer 1 Report

The original paper: "Allium derived compound propyl propane thiosulfonate (PTSO) attenuates metabolic syndrome in high fat diet-fed mice through its anti-inflammatory and prebiotic properties," presents an interesting but already known problem of the influence of sulfur compounds present in Allium spp on metabolic disorders. In mice, oral supplementation PTSO was found to reduce weight gain and improved plasma markers related to glucose and lipid metabolism, and reduce inflammation developing in diabetes. In my opinion, the manuscript is well prepared. Below I present my suggestions, which will help to improve this manuscript.

Abstract

- I propose to refine the purpose in the abstract a bit more. In my opinion, it is a bit too general.

Keywords

- Keywords should be different from the words used in the title - using other phrases will help you better perceive the article.

Introduction

- In the Introduction, the Authors describe the beneficial biological/pharmacological action of Allium, they briefly point to the active compounds of the Alliaceae family - I would ask you to include a brief presentation of biological activity "PTSO" described in the literature

- Please review the introduction for the rightly placed abbreviations (if the abbreviation does not appear in the text of the manuscript, it is not necessary to enter it, e.g., OSCs), as well as explain those whose meaning has not been given (e.g., UCP-1, EQF). This remark applies to the entire article (e.g., line 407 - mTOR - used only here)

- The authors should extend the description of the aim. It is worth indicate which kind of research will be performed to achieve the assumptions.

Results and Discussion

- I would suggest you divided this extensive paragraph into small parts, which should be titled. This passage is too rich in various data - it presents both the results and the discussion. The division will facilitate the receipt of work.

- Line 176 - PTSO has explained again - this is not necessary as the abbreviation has already been expanded in "Introduction" (line 80).

- Line 413-414 - the chart is incorrectly quoted and suggests that alline data can be found on it, while this one appears, I suppose, only to discuss the results received by the Authors.

Figures

- descriptions under the figures should explain the abbreviation PTSO

- The graphs in Figures (or the font size in the axis descriptions) should be slightly enlarged - they are not visible enough;

- individual graphs are not of equal size (Figure 5B - "Lyc6C + CD11b + (liver)")

- Figure 7B  and 7C - standardize the font in the legend of bacterial strains

- Figures 1 and 2 should be one below the other, while the fragment of the text separating them (line 209-214) would be better attached to the content placed before these two figures (after line 203).

- Figure 2A "HOMAR-IR" - "AUC" should be the y axis description and not part of the title (similar to Figure 1C)

- Figure 2 "HOMAR-IR" should be defined in the Figure caption

- Figure 7 - we can see "P <0.05" instead of "p <0.05"

Author Response

REVIEWER#1

Comments and Suggestions for Authors

The original paper: "Allium derived compound propyl propane thiosulfonate (PTSO) attenuates metabolic syndrome in high fat diet-fed mice through its anti-inflammatory and prebiotic properties," presents an interesting but already known problem of the influence of sulfur compounds present in Allium spp on metabolic disorders. In mice, oral supplementation PTSO was found to reduce weight gain and improved plasma markers related to glucose and lipid metabolism, and reduce inflammation developing in diabetes. In my opinion, the manuscript is well prepared. Below I present my suggestions, which will help to improve this manuscript.

 Abstract

- I propose to refine the purpose in the abstract a bit more. In my opinion, it is a bit too general.

The abstract has been modified according to the suggestion of the reviewer. 

Keywords

- Keywords should be different from the words used in the title - using other phrases will help you better perceive the article.

Following the helpful suggestion of the reviewer, the keywords have been changed.

“Keywords: cytokines; dysbiosis; glucose metabolism; lipid metabolism; microbiota; obesity; organosulfur compound”

 Introduction

- In the Introduction, the Authors describe the beneficial biological/pharmacological action of Allium, they briefly point to the active compounds of the Alliaceae family - I would ask you to include a brief presentation of biological activity "PTSO" described in the literature.

A brief description of the reported properties of PTSO has been included in the Introduction section:

In this sense, we evaluated an Allium organosulfur compound, propyl propane thiosulfonate (PTSO) in an experimental model of high fat diet-induced metabolic syndrome in mice. This product does not present toxic effects [18], and has already shown interesting antimicrobial properties for the livestock industry [19], and immunomodulatory effects in experimental colitis [20]. Thus, concerning the latter, it has shown to be able to down-regulate pro-inflammatory mediators, improve the intestinal mucosal integrity and ameliorate the colitis-associated dysbiosis [20]. These effects could also contribute to the amelioration of the metabolic syndrome-associated inflammation and its derived adverse sequels. The present study has assessed the effects of PTSO on the metabolic alterations and the inflammation that characterizes the metabolic syndrome, as well as on its abilities to modulate the gut microbiota.”

- Please review the introduction for the rightly placed abbreviations (if the abbreviation does not appear in the text of the manuscript, it is not necessary to enter it, e.g., OSCs), as well as explain those whose meaning has not been given (e.g., UCP-1, EQF). This remark applies to the entire article (e.g., line 407 - mTOR - used only here)

We are grateful for the comment and we have checked the manuscript to avoid misplacement of abbreviations.

- The authors should extend the description of the aim. It is worth indicate which kind of research will be performed to achieve the assumptions.

Following the reviewer’s suggestion, the aim of the study has been modified. 

Results and Discussion

- I would suggest you divided this extensive paragraph into small parts, which should be titled. This passage is too rich in various data - it presents both the results and the discussion. The division will facilitate the receipt of work.

We appreciate the reviewer’s comment and agree with it. We have divided the Results and Discussion Section introducing different headings

- Line 176 - PTSO has explained again - this is not necessary as the abbreviation has already been expanded in "Introduction" (line 80).

The change has been made.

- Line 413-414 - the chart is incorrectly quoted and suggests that alline data can be found on it, while this one appears, I suppose, only to discuss the results received by the Authors.

We agree with the reviewer and apologized for the mistake; the figure reference has been moved.

Figures

- descriptions under the figures should explain the abbreviation PTSO

- The graphs in Figures (or the font size in the axis descriptions) should be slightly enlarged - they are not visible enough;

- individual graphs are not of equal size (Figure 5B - "Lyc6C + CD11b + (liver)")

- Figure 7B  and 7C - standardize the font in the legend of bacterial strains

- Figures 1 and 2 should be one below the other, while the fragment of the text separating them (line 209-214) would be better attached to the content placed before these two figures (after line 203).

- Figure 2A "HOMAR-IR" - "AUC" should be the y axis description and not part of the title (similar to Figure 1C)

- Figure 2 "HOMAR-IR" should be defined in the Figure caption

- Figure 7 - we can see "P <0.05" instead of "p <0.05"

According to the reviewer suggestion, all figure modifications have been made.

Reviewer 2 Report

This manuscript is well written. The authors carried out various measures to evaluate the effects of Allium derived propyl propane thiosulfonate (PTSO) on the metabolic health of high fat diet induced obese mice. Minor changes are needed before being accepted for publication.

  1. Please justify why standard chow diet was chosen as the control instead of a low fat purified diet, which is considered as a better control for high fat purified diet.
  2. The use of a,b,c to indicate statistical significance in all the figures needs to be corrected. It needs to follow a>b>c or a<b<c, but the authors used b>a and c<b. In Figures 3 and 5, there is no letter for several bars.
  3. References are needed for the first several sentences in Introduction.
  4. Since the term "metabolic syndrome" has its clear definition and it's used for human, I would suggest the authors to use "metabolic perturbations or disturbance" in the title instead of "metabolic syndrome". 

Author Response

This manuscript is well written. The authors carried out various measures to evaluate the effects of Allium derived propyl propane thiosulfonate (PTSO) on the metabolic health of high fat diet induced obese mice. Minor changes are needed before being accepted for publication.

  1. Please justify why standard chow diet was chosen as the control instead of a low fat purified diet, which is considered as a better control for high fat purified diet.

There is no consensus for the control diet used in this experimental model and different studies use the standard diet as a control as well as the low fat diet (Lang et al. Sci Rep 9, 19556 (2019). https://doi.org/10.1038/s41598-019-55987-x; Wang et al. Life Sci. 2017 Dec 15;191:122-131. doi: 10.1016/j.lfs.2017.08.023.; Yashiro et al. Diabetes Obes Metab. 2019 Oct;21(10):2228-2239. doi: 10.1111/dom.13799.; Diez-Echave et al. Food Res Int. 2020 Jan;127:108722. doi: 10.1016/j.foodres.2019.108722.). Nevertheless, the low fat diet has quite similar contents of fat and carbohydrates to the standard chow that we have used in the present study: 6% fat and 40% carbohydrates vs 5.5% fat and 55.4%.

  1. The use of a,b,c to indicate statistical significance in all the figures needs to be corrected. It needs to follow a>b>c or a<b<c, but the authors used b>a and c<b. In Figures 3 and 5, there is no letter for several bars.

We apologize for the mistakes and we have revised the use of the letters to show the significant differences among  groups, as it is explained in the figure legends.

  1. References are needed for the first several sentences in Introduction.

Following the comment of the reviewer, several references have been included.

  1. Since the term "metabolic syndrome" has its clear definition and it's used for human, I would suggest the authors to use "metabolic perturbations or disturbance" in the title instead of "metabolic syndrome".

As the reviewer suggests, the title has been modified, and the term metabolic syndrome has been removed:

“Allium derived compound propyl propane thiosulfonate (PTSO) attenuates metabolic alterations in high fat diet-fed mice through its anti-inflammatory and prebiotic properties”

Reviewer 3 Report

This manuscript “Allium derived compound propyl propane thiosulfonate (PTSO) attenuates metabolic syndrome in high fat diet-fed mice through its anti-inflammatory and prebiotic properties”. The comments for this manuscript are as follows:

  1. The most important part of this manuscript that has research value is to conduct animal experiments, which is the advantage of this manuscript. However, the experimentally regrettable part is that the authors stated in the abstract that PTSO improved the expression of pro-inflammatory cytokines, but in fact it only measured the gene expression of these inflammatory proteins, not the expression content of these inflammatory proteins. There is a big difference between the two items. The genetic changes of these inflammatory proteins may not necessarily show up in the blood or liver. Only the determination of protein expression is a more meaningful study, and gene expression can only be used as a reference. Moreover, referring to the literature published by the authors in 2019, it is unreasonable to know why the animal experiment has been done, but they return to only determine the gene expression. This is very unreasonable. I strongly request that the authors should measure the amount of changes in the content of these inflammatory proteins, not the amount of changes in their genes. I think that these blood samples should be kept in laboratory, so the authors only needs to re- determine the amount of protein change is enough. I believe this should not be difficult for the authors, because they have already done it in the article published in 2019. Otherwise, I personally think that the amount of genetic change alone is not enough to prove that PTSO is anti-inflammatory effect.

Reference:

Teresa Vezzaa et al., 2019 (The immunomodulatory properties of propyl-propane thiosulfonate contribute to its intestinal anti-inflammatory effect in experimental colitis, Mol. Nutr. Food Res.)

  1. All inflammatory proteins should be written in the same way. For example, IL-1β, TNF-α, IL-6, etc., should be capitalized. It is rare to see Il-1β, Tnf-α. Because of those are the abbreviations of some words, how can they be lower cased? This is basic common sense, right?
  2. Please unify the format of the references. Do not use capital letters at the beginning of each word in the reference title. For example the page 15, reference 2, 3, 4, 5, 8, 14, page 16, reference 25 , 26, 29, 32, page 17, reference 39, 50, 60, 62 and page 18, reference 69, etc.
  3. All scientific names should be in italics, please recheck all of the manuscript and correct it. For example, reference 19, 21... etc.

Author Response

REVIEWER#3

Comments and Suggestions for Authors

This manuscript “Allium derived compound propyl propane thiosulfonate (PTSO) attenuates metabolic syndrome in high fat diet-fed mice through its anti-inflammatory and prebiotic properties”. The comments for this manuscript are as follows:

  1. The most important part of this manuscript that has research value is to conduct animal experiments, which is the advantage of this manuscript. However, the experimentally regrettable part is that the authors stated in the abstract that PTSO improved the expression of pro-inflammatory cytokines, but in fact it only measured the gene expression of these inflammatory proteins, not the expression content of these inflammatory proteins. There is a big difference between the two items. The genetic changes of these inflammatory proteins may not necessarily show up in the blood or liver. Only the determination of protein expression is a more meaningful study, and gene expression can only be used as a reference. Moreover, referring to the literature published by the authors in 2019, it is unreasonable to know why the animal experiment has been done, but they return to only determine the gene expression. This is very unreasonable. I strongly request that the authors should measure the amount of changes in the content of these inflammatory proteins, not the amount of changes in their genes. I think that these blood samples should be kept in laboratory, so the authors only needs to re- determine the amount of protein change is enough. I believe this should not be difficult for the authors, because they have already done it in the article published in 2019. Otherwise, I personally think that the amount of genetic change alone is not enough to prove that PTSO is anti-inflammatory effect.

Reference:

Teresa Vezzaa et al., 2019 (The immunomodulatory properties of propyl-propane thiosulfonate contribute to its intestinal anti-inflammatory effect in experimental colitis, Mol. Nutr. Food Res.)

We agree with the reviewer and think that it is important to check that the changes in gene expression correlate with changes in protein expression. In this case, we have used all the blood samples for the biochemical determinations and we could not carry out the protein determinations that the reviewer suggests. However, we consider that we have proven before that there is a correlation between the gene expression and the protein expression when the effects of PTSO were evaluated in experimental models of colitis, so we thought that it was not ethical to repeat the mouse model in order to carry out those measurements.

  1. All inflammatory proteins should be written in the same way. For example, IL-1β, TNF-α, IL-6, etc., should be capitalized. It is rare to see Il-1β, Tnf-α. Because of those are the abbreviations of some words, how can they be lower cased? This is basic common sense, right?

We apologized for the mistake. We have revised the manuscript to write properly the expression of genes (italicized, with only the first letter in upper-case) and proteins (not italicized, and all letters are in upper-case).

  1. Please unify the format of the references. Do not use capital letters at the beginning of each word in the reference title. For example the page 15, reference 2, 3, 4, 5, 8, 14, page 16, reference 25 , 26, 29, 32, page 17, reference 39, 50, 60, 62 and page 18, reference 69, etc.

We have thoroughly revised the references.

  1. All scientific names should be in italics, please recheck all of the manuscript and correct it. For example, reference 19, 21... etc.

We again apologize for the mistake and we have revised the manuscript to correct the inconsistencies.

Round 2

Reviewer 3 Report

For such a topic, I think that I can accept this manuscript.